# HeightNet: Monocular Object Height Estimation

**In Su Kim** [1] , **Hyeongbok Kim** [2] **, Seungwon Lee** [2] **and Soon Ki Jung** [1,*]

1  School of Computer Science and Engineering, Kyungpook National University, Daegu 41566, Republic of Korea
2  Testworks, Inc., Seoul 01000, Republic of Korea
*  Correspondence: skjung@knu.ac.kr

**Abstract:** Monocular depth estimation is a traditional computer vision task that predicts the distance of each pixel relative to the camera from one 2D image. Relative height information about objects lying on a ground plane can be calculated through several processing steps from the depth image. In this paper, we propose a height estimation method for directly predicting the height of objects from a 2D image. The proposed method utilizes an encoder-decoder network for pixel-wise dense prediction based on height consistency. We used the CARLA simulator to generate 40,000 training datasets from different positions in five areas within the simulator. The experimental results show that the object's height map can be estimated regardless of the camera's location.

**Keywords:** object height estimation; virtual dataset; deep learning

## 1. Introduction

Recently, deep learning techniques have been actively researched in various directions to estimate useful information from monocular images. In addition to general object detection or tracking, it has also been applied to domains such as depth estimation or object segmentation that require 3D information. One of the deep learning-based studies is an encoder-decoder network that uses monocular images as inputs to extract features from images and then generate them as suitable images for processing while recreating them. This encoder-decoder network has been widely studied as an architecture mainly used in domains requiring pixel-wise prediction.

Depth estimation, one of the pixel-wise predictions, is a traditional problem in computer vision for extracting geometric information of a scene from a single image; deep learning-based approaches to solving this problem yield excellent results in limited environments [1,2]. Adabins is one of the supervised learning models with an encoder-decoder network for monocular depth estimation. Unlike other methods approaching the regression problem, the authors discretized the depth values and estimated them effectively by approaching the classification problem [3]. Lee et al. created a feature map using dense feature extractor and extracted contextual information using atrous spatial pyramid pooling (ASPP). In addition, the authors proposed a method to effectively reconstruct the resolution back to the input image size by utilizing the local planar guidance layer to identify the large scale variation [4].

Height estimation is similar to depth estimation in extracting geometric information from images, but it estimates the vertical length in terms of the dominant reference plane within the scene. Height estimation has various applications, such as health checks for patients [5], growth checks for plants or animals [6], 3d building modeling and change detection [7], semantic labeling of objects in aerial images [8,9], and obstacle detection [10]. In the case of aerial images taken from high positions, height estimation is the same problem as depth estimation. However, if the camera is taken obliquely from the reference ground, the two problems are quite different. The estimation of height from the depth image

requires additional processing steps, such as estimating the ground plane and measuring the vertical distance of the reference plane.

This paper proposes a height estimation method for directly predicting pixel-wise dense height maps of objects from a 2D image. The proposed method exploits the height consistency of the moving objects observed in a fixed camera to extract consistent information from the object's unique feature vectors according to the characteristics that their height values remain constant regardless of the relative position of the camera. The encoder-decoder architecture of the state-of-the-art depth estimation models can easily apply to the height estimation based on height consistency. It enables the efficient utilization of robust encoder-decoder models, thereby improving the performance of height estimation models. We also propose a synthetic dataset with a virtual simulator for training height estimation models and use this dataset to show experimental results.

## 2. Related Works

### 2.1. Encoder-Decoder Networks

Encoder-decoder networks have made significant contributions to many computer vision-related problems, such as image segmentation, image optical flow, image reconstruction, and image analysis [3,11]. In recent years, the use of such architectures has demonstrated great success in both supervised and unsupervised learning for depth estimation problems [12,13]. These methods generally produce good results by utilizing one or more encoder-decoder networks in various ways. It is used as a submodel of a particular architecture or as a parallel or series connection depending on the domain or application method. Furthermore, many studies have shown that using additional skip connections concurrently, state-of-the-art high-quality image segmentation, image generation, and image reconstruction results can be achieved in encoder-decoder architectures.

### 2.2. Height Estimation

Most height estimation methods have been proposed for aerial photographs or orthoimages from satellites rather than ground images taken from CCTV or other perspective views. Various deep learning methods for height estimation from a single monocular remote-sensing image have been proposed. Srivastava et al. proposed the joint prediction of height and semantic labels in a multitasking deep learning framework [8]. Mou et al. designed residual convolutional neural networks for height estimation and demonstrated their effectiveness on instance segmentation [9]. Furthermore, a conditional generative adversarial network was proposed for framing height estimation as an image-translation task. H. Dami et al. proposed a method for estimating crop heights on point cloud maps using 3D LiDAR sensors in unmanned aerial vehicles [6]. Anua Trivedi et al. proposed a CNN-based approach to estimate the height of a standing child under the age of 5 from depth images collected using a smartphone [5]. Kunwar et al. exploited semantic labels as priors to enhance the height estimation performance on the semantic dataset [14]. Xiong et al. designed and constructed a large-scale benchmark dataset for cross-layer transfer learning for height estimation tasks [15], including large-scale synthetic and real-world datasets. Previous height estimation studies have limitations in that the viewpoints are limited, or additional sensors such as Lidar are required. Therefore, this paper proposes a deep learning-based height estimation method without limitations on the camera position.

### 2.3. Pixel-Wise Dense Prediction

Pixel-wise dense prediction is a problem of calculating the value of each pixel in an image by analyzing the feature of the environment. Previous work estimated pixel-wise values using environmental models, geometric features, or mathematical formulas (Lambert models, triangulations, etc.). However, in recent years, most studies have applied deep learning to derive good performance. Dijk et al. investigated important visual cues in monocular images for depth estimation [16]. Hu et al. attempted to determine the most relevant sparse pixels for depth estimation [17]. You et al. first determined

the depth selectivity of some hidden units, which showed good insights for interpreting monocular depth estimation models [18]. Zhi et al. proposed a novel disentangled latent Transformer model based on the multi-level interpretation [19]. Santo et al. proposed a deep photometric stereo network (DPSN) by applying a photometric stereo method based on deep learning away from simplified image formation models such as the general Lambert model [20]. Ju et al. proposed to solve the problem of blurred reconstruction due to the limitations of methods using deep learning and to improve surface direction prediction for complex structures [21]. Logothetis et al. proposed a PX-NET network that utilizes data as independent data generated per pixel to perform new pixel-wise training for normal prediction [22].

### 2.4. CARLA Simulator

The dataset for height estimation is difficult to obtain. Therefore, recently, methods of utilizing simulators in domains where it is challenging to build datasets have been studied. In this study, a dataset was created using a CARLA simulator [23]. CARLA is an open-source simulator that is used in autonomous driving research. CARLA has been developed from the ground to support autonomous urban driving systems' development, training, and validation. In addition to open-source code and protocols, CARLA provides open digital assets (urban layouts, buildings, and vehicles) that have been created for this purpose and can be used freely. The simulation platform supports flexible specifications of sensor suites and environmental conditions. In this study, a dataset was manufactured using the CARLA simulator for cameras at various positions and heights of the map provided for estimating the height.

The proposed height estimation method estimates the height from the ground for each object in the image to extract more spatial information using the image input from the monocular camera.

### 3. Methods

Our proposed object height estimation model aims to estimate the height of each pixel from the ground through an encoder–decoder network. The estimated height can be converted into real-world height when the height of the camera can be calculated, which is suitable for use in various fields.

### 3.1. Height Map Estimation Using Encoder-Decoder Network

The overall architecture of the proposed method is shown in Figure 1. The network was configured as a pre-trained encoder and decoder to estimate the object's height. The encoder efficiently uses various networks, such as VGG [24], EfficientNet [25], and ResNeXt [26], depending on the environment or the characteristics of applying the model by actively utilizing the architecture provided by the deep learning framework. The encoder can extract more detailed and essential features from the input image with an increase in the depth of the CNN layer. Although the spatial size of these feature maps becomes very small, they comprehensively include the relationship between the color textures and height values of the inner space learned in various scene geometries. This paper used LapDepth [27], which utilizes edge information for height estimation, as the underlying architecture. This architecture proposed to adopt the Laplacian pyramid to solve the monocular camera-based estimation problem. This adopts the DenseASPP technology to obtain context information more densely and with basic encoders. This is a highly effective method for extracting more detailed information about the pixels of an image. The decoder uses bilinear upsampling and a skip-connection with edge information to generate a dense feature map extracted from ASPP. This proposed method adopted weight standardization for the pre-activation convolution blocks for stable network training. By combining this spatial information and global information, accurate final height map results are produced.

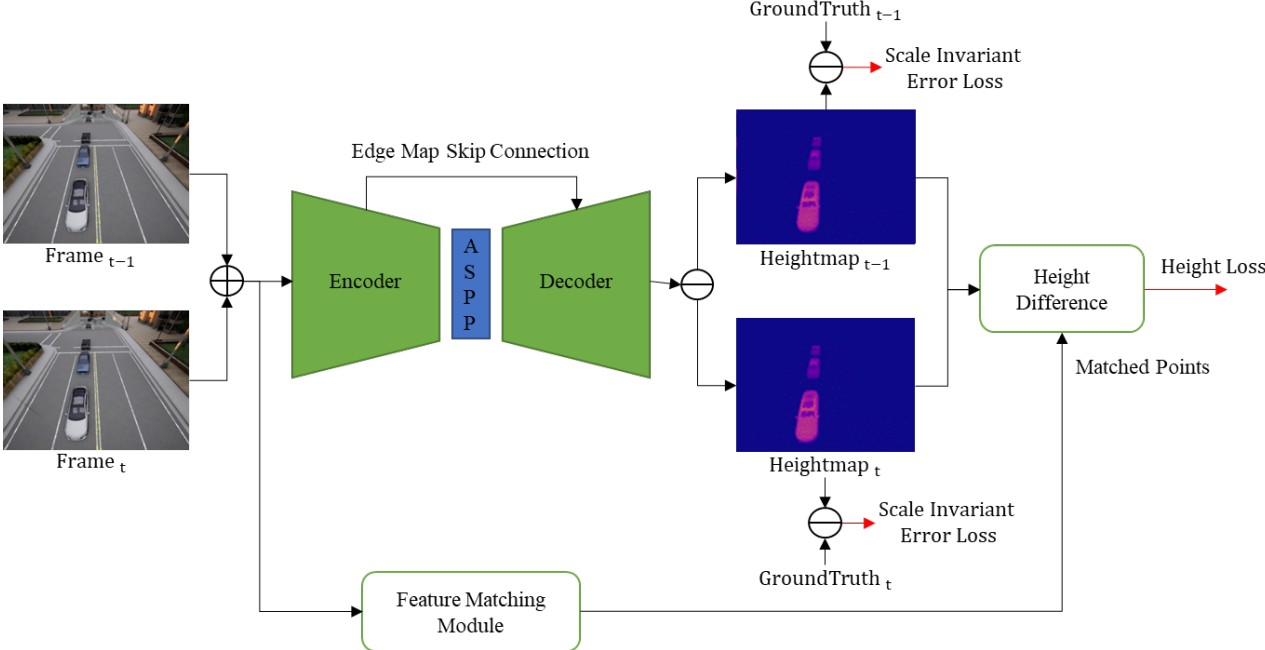

**Figure 1.** Overview of HeightNet (Height Map Estimation Network).

### 3.2. Ground Plane Segmentation

To accurately estimate the height of an object, it is necessary first to set the reference accurately. When estimating depth, the reference value becomes the camera's current location. The depth values in the image change according to the camera position. However, height estimation is different from depth estimation. Even if the position of the camera changes, the exact height value must be continuously maintained. Accordingly, the reference is set not to the camera but to the ground plane in the current scene. To determine the height based on the ground, it is necessary to distinguish between the background and foreground. Therefore, the value of the ground was set to 1 for pixel values between 0 and 255, and the value increased as it rose higher from the ground based on the value. It learns how to distinguish between the ground plane and the objects by estimating the height. This is important for height estimation based on the distance from the ground plane. Thus, the objects to be learned can be separated from the background so that the encoder and decoder can obtain accurate results from the input image. When estimating the height of an object by increasing the effectiveness of the background estimation, better results can be obtained by specifying the location of the ground value of the height through the division with the object and stacking the height from that location.

### 3.3. Height Consistency of Objects and Loss Function

Unlike depth estimation, the height of a moving rigid object does not change, regardless of its location in the image. The height consistency is an essential clue for height estimation, and its active use can significantly improve accuracy. In this paper, we estimated feature correspondences in the multiple frames taken from the fixed camera and analyze whether the calculated height of the corresponding feature point has the same value, as shown in Figure 1.

### 3.4. Loss Function and Dataset

The proposed method utilizes two types of loss functions. The first is the scale-invariant error loss [28] used by the height-estimating encoder-decoder network. Scale-invariant data loss, which is widely used in depth estimation, can also be used for height

estimation, where it computes the difference between the predicted height values and the ground truth in log space as follows:

$$L_s(y, y^*) = \sqrt{\frac{1}{n}\sum_i h_i^2 - \frac{\lambda}{n^2}\left(\sum_i h_i\right)^2} \,, \tag{1}$$

where $h_i = log(y_i) - log(y_i^*)$ is the difference between the prediction and ground truth at pixel $i$, and $n$ is the total number of pixels. The balancing factor $\lambda$ is set to 0.85 in the same way used in [27].

Second, we designed a loss using the difference in heights of the matched features of the two images and utilized it for learning the encoder-decoder networks. This height consistency loss is calculated by summing the differences between all matched features as follows:

$$L_h(y^{Frame_t}, y^{Frame_{t-1}}) = \frac{1}{n}\sum_i^n (y_i^{Frame_t} - y_i^{Frame_{t-1}})^2 \,, \tag{2}$$

where $y_i^{Frame_t}$ and $y_i^{Frame_{t-1}}$ denote the height value of pixel $i$ in $Frame_t$ and $Frame_{t-1}$, and $n$ is the total number of pixels.

This study generated a dataset by taking three videos of moving-object scenes, RGB, depth, and segmentation, from several fixed cameras in the CARLA simulator. The ground plane was segmented using the segmentation image. As shown in Equation (3), using the height of a camera and the depth map, the height of the object was determined through a triangular proportional expression. In addition, height estimation could be made based on the depth map without directly extracting the height, allowing the ground truth generation for height estimation using various open datasets. The formula used was:

$$H_i = \frac{C_h * D_i^{Background}}{D_i^{Background} - D_i^{Foreground}} \,, \tag{3}$$

where $H_i$ and $C_h$ denote the height value of pixel $i$ in the image and the height value of camera in image. $D_i^{Background}$ and $D_i^{Foreground}$ is the depth value of pixel $i$ in background and foreground images.

Therefore, to obtain the ground truth of height, a background image of the corresponding scene is required. The depth value of the background is the ground value of the corresponding pixel. In addition, because the foreground pixel value is closer to the camera than the ground, it could be used as a value for height. To obtain a more accurate height value, it was possible to calculate how high the estimated height is by substituting the actual height of the camera.

## 4. Experimental Results

This section describes the dataset generated by CARLA along with the learning details. We also show that the estimated height calculated using the height estimation learned from the described information is sufficiently reliable. In addition, an ablation study was performed for height loss to show significant results.

### 4.1. Dataset and Training Details

In this study, for height map estimation, a dataset was generated using the CARLA simulator. The dataset was created by photographing scenes in which time or weather changed naturally in the simulator based on the height of five different cameras in five scenes. Some of the sample images are shown in Figure 2. And every image has the groundtruth image as shown in Figure 3. A total of 44,000 data were generated, divided into approximately 40,000 learning data and approximately 4000 test data, and used for learning.

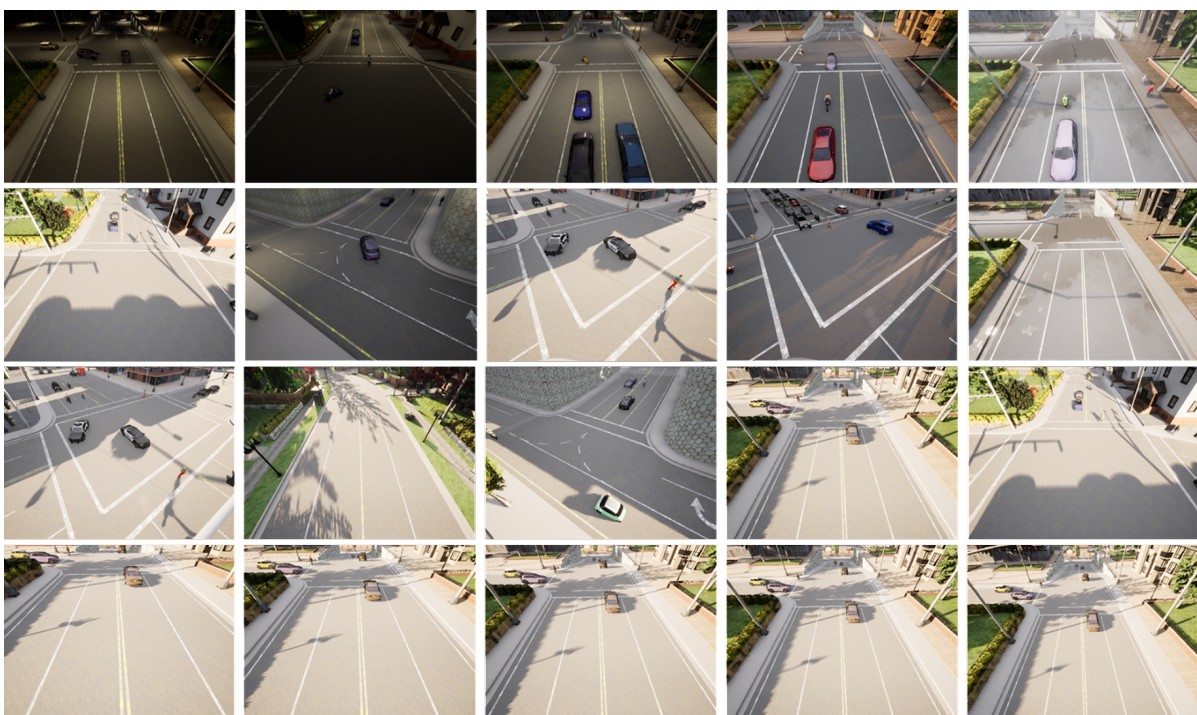

**Figure 2.** Images generated from CARLA simulator [23] with different weather conditions (**1st row**), different times of the day (**2nd row**), different camera locations (**3th row**), and different camera heights (**4th row**).

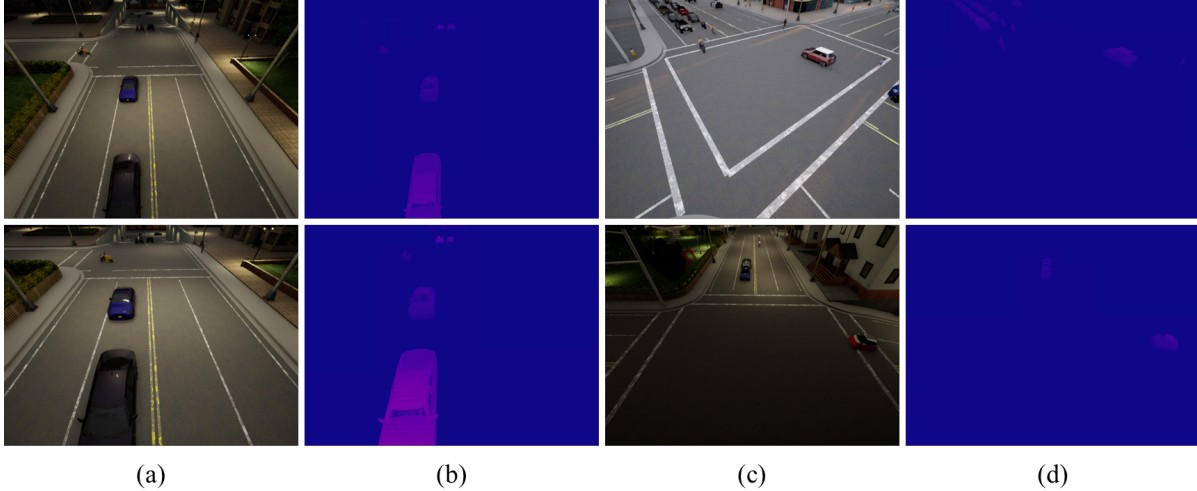

(a)       (b)       (c)       (d)

**Figure 3.** Generated height dataset using CARLA Simulator [23]. (**a,c**) represents the input color images, (**b,d**) is Ground Truth of each input color images.

All models were trained for 20 epochs with a batch size of 2. We also applied various augmentations. For the baseline model, we followed the LapDepth model, with the ResNext encoder and a decoder, which consisted of a bilinear upsampling system. All experiments were conducted using the free distribution of Anaconda with Python 3.7. The models were implemented using the Pytorch library as a deep learning framework. We used four NVIDIA TITAN xp GPUs with 12 GB RAM for computation.

*4.2. Results*

To demonstrate the effectiveness of the proposed network, experiments were conducted through several methods.

Table 1 shows the results of quantitative experiments on datasets generated by the CARLA simulator. The experiment was conducted by dividing the experimental data into five camera heights. In the experimental results, as the height of the camera increases, it can be shown that the error rate slightly increases as more objects are detected. However, the error rate does not differ much compared to the area of the search area. It may be seen that the height estimation method is not greatly affected by the position of the camera.

The results of our proposed framework is shown in Figure 4. Parts close to the horizontal line corresponding to the distance in the input image were excluded from the estimation. The height map represents a height value in the range of 1–255 from the floor. The darker the color, the higher the value. In each image, it can be observed that the height value appears regardless of the location of the camera. It can also be confirmed that the height can be estimated equally in the installation environment or under changing weather conditions. This result is possible because the method considers changes in the height, location, and environment of the camera in the learning data.

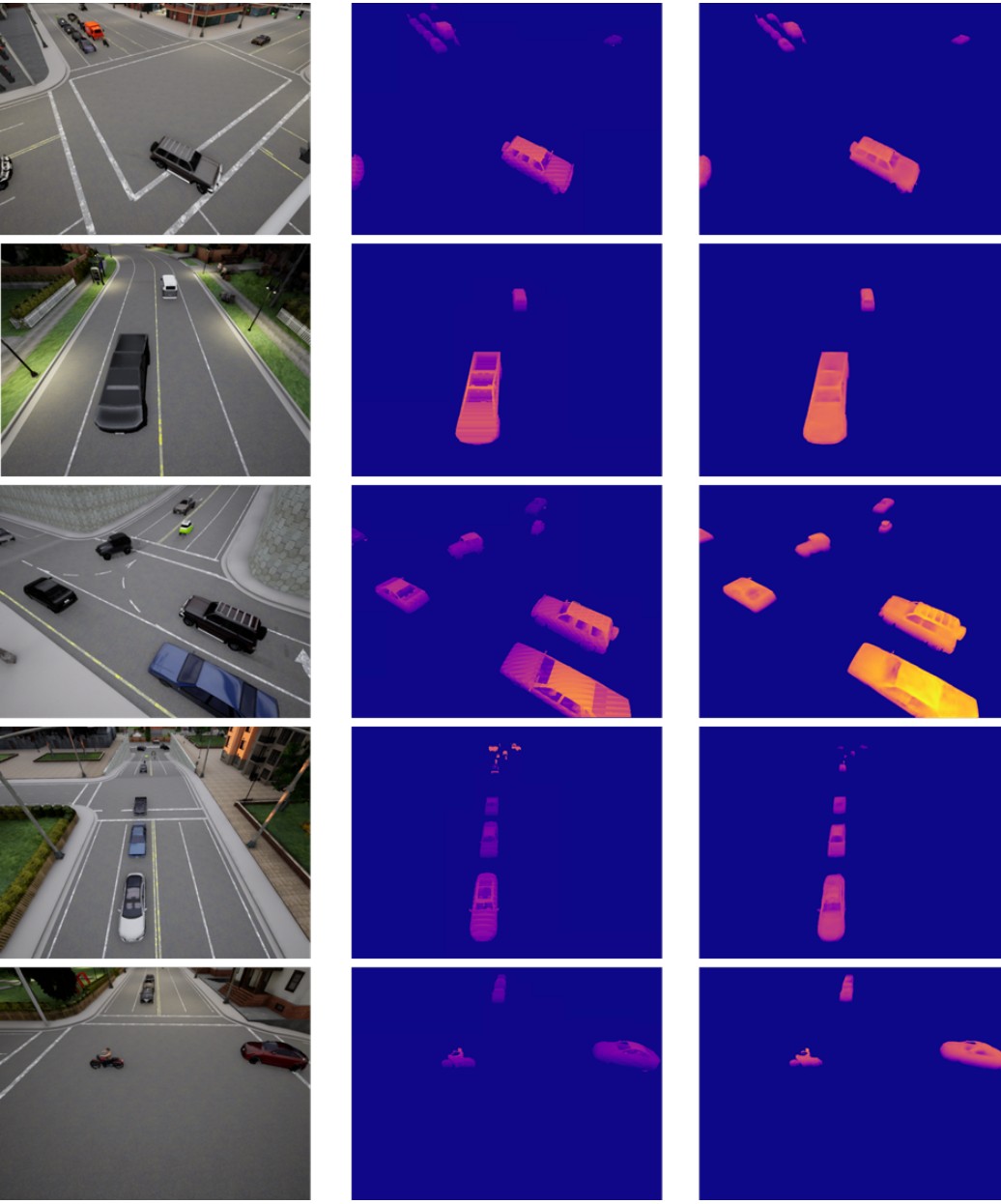

**Figure 4.** Experimental Results of Height Estimation Network. Original image (**Left**), Ground Truth (**Middle**), Estimated height (**Right**).

**Table 1.** Height Estimation Results on generated height dataset using CARLA simulator.

| Camera Height (cm) | RMSE | $\delta = 1.25$ | $\delta = 1.25^2$ | $\delta = 1.25^3$ |
|---|---|---|---|---|
| 350 | 3.286 | 1.113 | 1.291 | 1.525 |
| 500 | 3.770 | 1.119 | 1.301 | 1.518 |
| 640 | 4.190 | 1.109 | 1.271 | 1.464 |
| 780 | 4.659 | 1.118 | 1.253 | 1.394 |
| 920 | 5.056 | 1.114 | 1.242 | 1.373 |

Figure 5 shows the results of learning by converting KITTI's Depth Dataset [29] as if it had generated a CARLA dataset. The KITTI Depth Dataset was collected through sensors attached to cars. As this is not a fixed-camera environment, the environment continues to change in real time. In addition, it is characteristically difficult to secure a dense pixel data value because the data in this dataset were collected using a sensor. This characteristic is illustrated in Figure 6. The results obtained using the KITTI dataset are very unstable compared with the results generated by the CARLA simulator. This is because of the nature of the dataset. For more accurate results, fixed environmental datasets such as CCTV are more advantageous.

Example of failure may occur if the object is too close to distinguish the object, if a part of the object disappears due to the specificity of the object, or the difference in the size of the object is not reflected in the learning data. Some of the examples are shown in Figure 7. These parts still need to supplement the dataset and further learn to produce stable results. In addition, stronger learning will be possible by creating a connection point through object detection for parts that follow, such as large trucks.

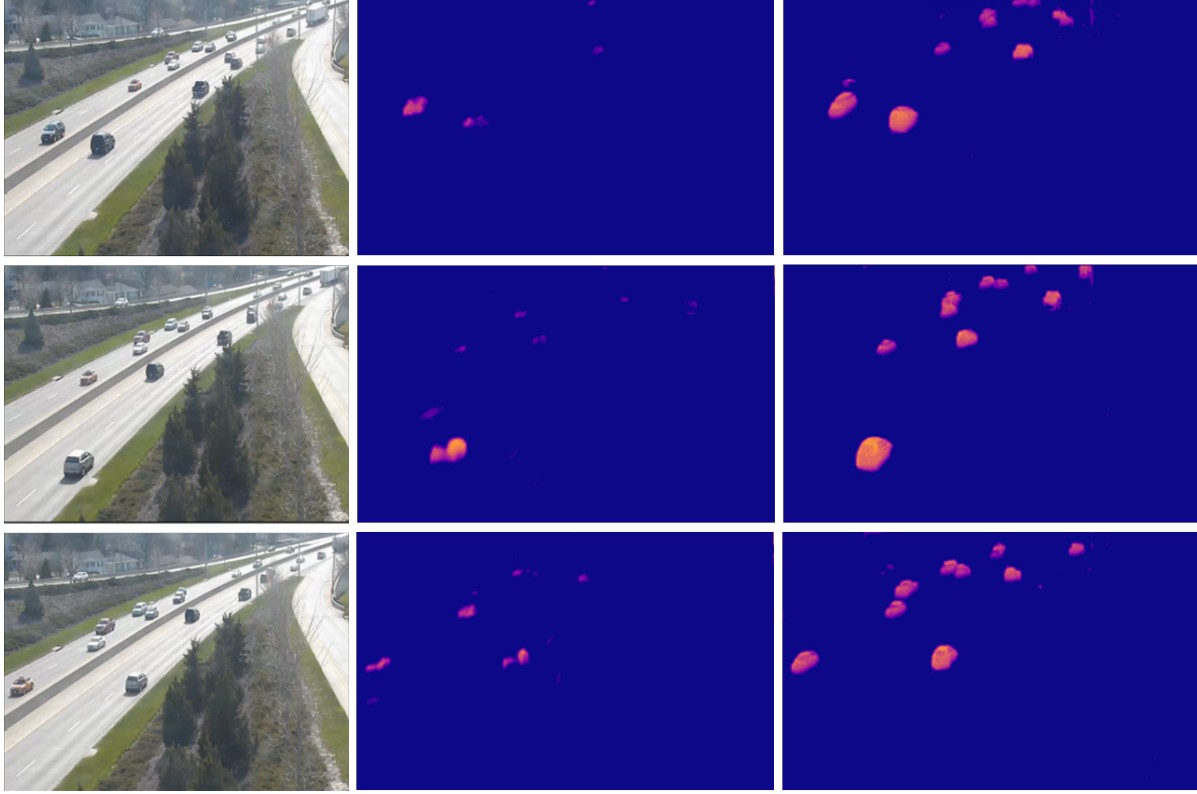

**Figure 5.** Original image (**Left**); Kitti-based results (**Middle**); CARLA-based results (**Right**).

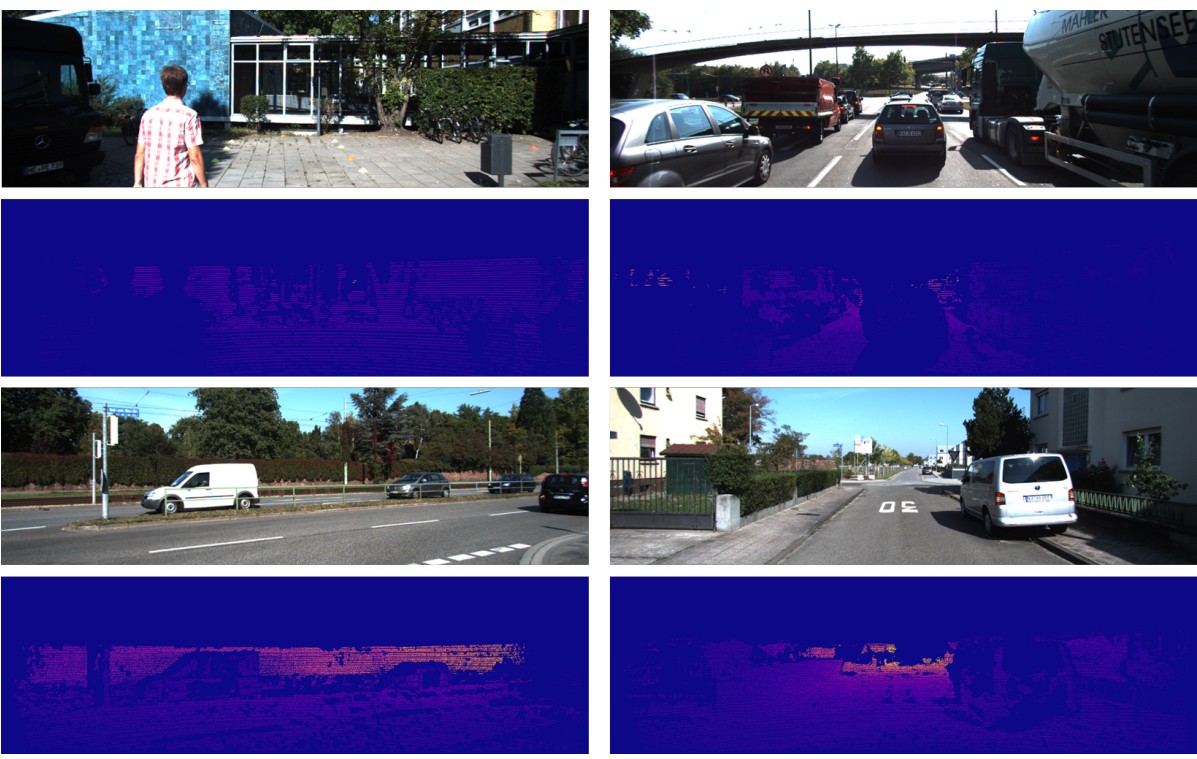

**Figure 6.** Original image (**1st row**, **3rd row**); generated Ground Truth (**2nd row**, **4th row**).

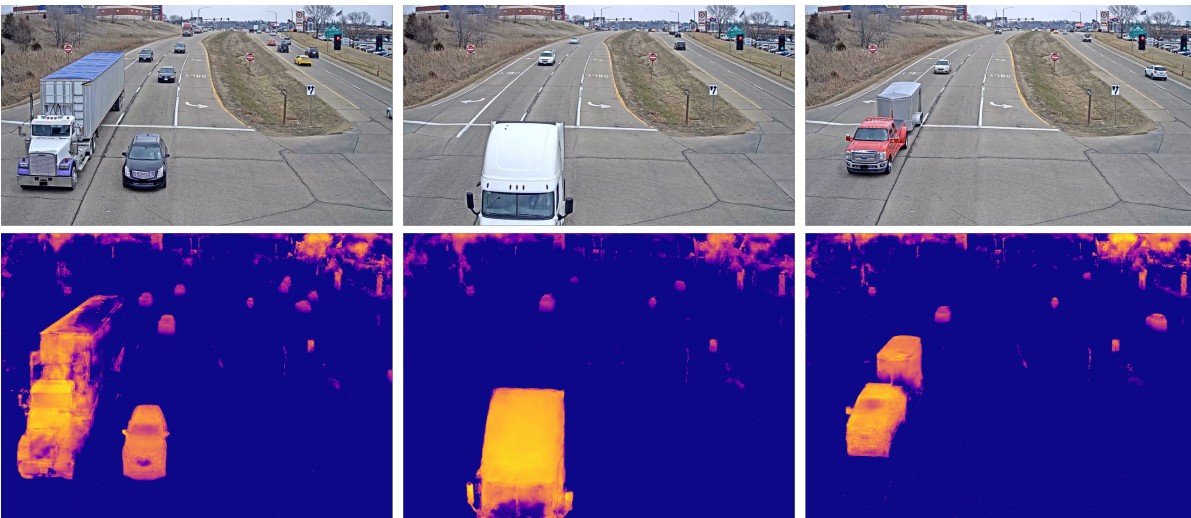

**Figure 7.** Failure cases of Height Estimation.

## 5. Conclusions

In this paper, we propose a method for height map estimation in monocular images to utilize 3D information. The main idea is to use the encoder–decoder network to generate the height map from the monocular image and to create a stronger height map using the height consistency. This can be used for the augmentation of an object by estimating the height as an image obtained from a monocular camera or applied to various fields necessary for extracting 3D information. We create a new height dataset using the CARLA simulator. The number of all datasets is about 50,000, which diversifies the weather and environment for better training results. The experimental results presented that the value changed as the object rose from the ground and estimated the height value regardless of the location of the camera. There is a limit to estimating the height of an object that still occupies a large area in the image or estimating the height of an object that is hard to see because the bottom of it

is covered. To this end, the network needs to study more robust height estimation methods, such as supplementing additional data and utilizing surrounding static objects.

**Author Contributions:** Conceptualization, I.S.K. and S.K.J.; methodology, I.S.K. and S.K.J.; software, I.S.K.; validation, I.S.K. and S.K.J.; formal analysis, I.S.K. and S.K.J.; investigation, H.K. and S.L.; resources, H.K. and S.L.; data curation, I.S.K.; writing—original draft preparation, I.S.K.; writing—review and editing, I.S.K. and S.K.J.; visualization, I.S.K.; supervision, S.K.J.; project administration, S.K.J.; funding acquisition, H.K. and S.L. All authors have read and agreed to the published version of the manuscript.

**Funding:** This study was supported by the BK21 FOUR project (AI-driven Convergence Software Education Research Program) funded by the Ministry of Education, School of Computer Science and Engineering, Kyungpook National University, Republic of Korea (4199990214394).

**Data Availability Statement:** All the dataset can be easily generated using CARLA [23]. It is an open-source simulator where you can generate environments with different conditions like weather, lighting, traffic, etc. Our method is performed on a private dataset which is generated using this simulator. The data are not publicly available but if anyone is interested, you can request from the corresponding author.

**Conflicts of Interest:** The authors declare no conflict of interest.

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
