# Peer review of "HeightNet: Monocular Object Height Estimation"

_electronics, doi:10.3390/electronics12020350_

Round 1
Reviewer 1 Report
The authors declare that this manuscript proposes a network that aims to estimate the height of an object in an image to derive three-dimensional information from a monocular image. Also, the authors create a new dataset.
Some major issues I want to address:
1. What is the application value of height estimation, compared with the depth estimation?
2. The 2nd paragraph in the Introduction is hard to understand. The authors say "Because there are not many studies" in height estimation, but the following content discusses depth estimation. . Please double-check the logic and purpose of this paragraph.
In fact, the goal and motivation of this manuscript are very low, as the currently written Abstract and Intro., please revise carefully. Do not use "However, the simple use of these ideas is insufficient." The author should discuss why these ideas are insufficient! Also, the contribution is no content, the authors should write what kind of network they design.
3. Many typos, such as: using et al. [1] rather than et al [1].
4. In line 75-77, "The height of an object is an inherent attribute that should not change under different views, whereas the depth of an object is highly dependent on the camera pose." I think this sentence is important and should be highlighted in the beginning, to let the readers know what the difference.
5. The literature only reports a few works from previous, not recent works. I only see 3 papers after 2021. The authors should improve the related work, and classify the novelty and difference between the proposed method and the previous.
6.The authors say "By estimating the height of an object, three-dimensional information can be obtained more accurately from a single image." However, in fig.4, the estimated height is very blurry without details, I think it hardly be used in 3D recovery.
7. The network architecture should be detailed shown, whether in Fig. 1 or using words. Even if the authors use the LapDepth without changes.
8. There is no compared method in the authors' experiments.
9. Similar to monocular depth/ height estimation, deep learning-based photometric stereo is another single-view method, which can be used in reconstructing the 3D information of an object. It would be nice to add a brief introduction with some references, such as:
[1] Santo H, Samejima M, Sugano Y, et al. Deep photometric stereo network[C]//Proceedings of the IEEE international conference on computer vision workshops. 2017: 501-509.
[2]Ju Y, Shi B, Jian M, et al. NormAttention-PSN: A High-frequency Region Enhanced Photometric Stereo Network with Normalized Attention[J]. International Journal of Computer Vision, 2022, 130(12): 3014-3034.
[3]Logothetis F, Budvytis I, Mecca R, et al. Px-net: Simple and efficient pixel-wise training of photometric stereo networks[C]//Proceedings of the IEEE/CVF International Conference on Computer Vision. 2021: 12757-12766.
Author Response
Thank you for your comment and suggestion.
Please see the attachment.

Reviewer 2 Report
Authors Have presented their work with the title “HeightNet: Monocular object height estimation”
The research work contributed by the authors of this paper is mainly reflected in the following points:
· Authors have worked on how to estimate the height of an object, for that they proposed a model that improves accuracy by utilizing objects in consecutive frames.
· Authors have also highlighted in the article i.e., by estimating the height of an object, three-dimensional information can be obtained more accurately from a single image.
· Authors have used the Carla simulators; with this they acquired the new height dataset and also the experimental results were trained from the simulator dataset.
· English language and style are fine/minor spell check required and check whether all the references have cited in the running text or not.
· I request the authors to avoid the words like I, WE, YOU in the paper running text and the paper should be written in third person format.
Author Response

(The authors gave the same response as above.)

Reviewer 3 Report
This paper proposes a monocular object height estimation method,it is an interesting topic. Some comments are as follows:
(1) The authors do not give the details of the proposed method, it can not support the reader to reproduce this wok, please give more details of the proposed method, every step need be given the more details.
(2) In the simulation, how to generate the the datasets, please give more details.
(3) In the comparison of the results , the reviewer suggests give the quantitative results.
Author Response

(The authors gave the same response as above.)

Round 2
Reviewer 1 Report
The authors have addressed all my concerns.
Reviewer 3 Report
All questions I raised have been addressed, thanks the authors for the careful revision.